# Integrative analysis of miRNA and mRNA sequencing data reveals potential regulatory mechanisms of ACE2 and TMPRSS2

Stepan Nersisyan[1,2]*, Maxim Shkurnikov[3], Andrey Turchinovich[4], Evgeny Knyazev[1,5], Alexander Tonevitsky[1,5]

**1** Faculty of Biology and Biotechnology, Higher School of Economics, Moscow, Russia, **2** Faculty of Mechanics and Mathematics, Lomonosov Moscow State University, Moscow, Russia, **3** P.A. Hertsen Moscow Oncology Research Center, Branch of National Medical Research Radiological Center, Ministry of Health of the Russian Federation, Moscow, Russia, **4** SciBerg e.Kfm, Mannheim, Germany, **5** Shemyakin-Ovchinnikov Institute of Bioorganic Chemistry RAS, Moscow, Russia

* s.a.nersisyan@gmail.com

## Abstract

Development of novel approaches for regulating the expression of angiotensin-converting enzyme 2 (ACE2) and transmembrane serine protease 2 (TMPRSS2) is becoming increasingly important within the context of the ongoing COVID-19 pandemic since these enzymes play a crucial role in cell infection. In this work we searched for putative ACE2 and TMPRSS2 expression regulation networks mediated by various miRNA isoforms (isomiR) across different human organs using publicly available paired miRNA/mRNA-sequencing data from The Cancer Genome Atlas (TCGA) project. As a result, we identified several miRNA families targeting ACE2 and TMPRSS2 genes in multiple tissues. In particular, we found that lysine-specific demethylase 5B (JARID1B), encoded by the KDM5B gene, can indirectly affect ACE2 / TMPRSS2 expression by repressing transcription of hsa-let-7e / hsa-mir-125a and hsa-mir-141 / hsa-miR-200 miRNA families which are targeting these genes.

## Introduction

The COVID-19 pandemic caused by the severe acute respiratory syndrome coronavirus 2 (SARS-CoV-2) had a dramatic impact on the health of millions of people and became a challenge for the worldwide health systems [1, 2]. High mortality rate of the infection is generally due to lung failure induced by the acute respiratory distress syndrome (ARDS) [3]. Furthermore, there are indications that SARS-CoV-2 can affect other organs including the digestive system [4], kidney [5] and brain [6]. The mechanism of host cell infection by SARS-CoV-2 is not completely understood and remains an active research topic. However, it became evident that the key players during the virus entry into the host cell are angiotensin-converting enzyme 2 (ACE2) and transmembrane serine protease 2 (TMPRSS2). Specifically, the interaction of

available on GitHub (https://github.com/s-a-nersisyan/miRNA_ACE2_TMPRSS2).

**Funding:** The article was prepared within the framework of the Basic Research Program at HSE University and funded by the Russian Academic Excellence Project '5-100'. The funder provided support in the form of salaries for authors A Tu, but did not have any additional role in the study design, data collection and analysis, decision to publish, or preparation of the manuscript. The specific roles of these authors are articulated in the 'author contributions' section.

**Competing interests:** Andrey Turchinovich is affiliated with the company SciBerg e.Kfm. This does not alter our adherence to PLOS ONE policies on sharing data and materials.

these membrane proteins with the viral spike protein (S-protein) is crucial for host-membrane fusion and endocytosis [7, 8].

MicroRNA (miRNA) is a class of small (about 22 nt long) non-coding RNAs, whose main function is negative post-transcriptional and translational regulation of gene expression [9]. In most cases, this regulation is mediated by complementary binding of miRNA to the 3' UTR of the target mRNA and promoting its degradation and/or translation repression [10, 11]. Furthermore, multiple reports have consistently shown that aberrant miRNA expression is associated with various pathologies including cancer [12–14], neurological [15, 16] and cardiovascular diseases [17, 18]. Finally, some host- and virus-encoded miRNAs can contribute to the process of infection by targeting certain mRNAs (both host and viral) [19, 20].

Previous analysis of miRNA sequencing reads showed that miRNAs are usually present in various isoforms (isomiRs) which differ from each other by 1-3 nucleotides at the 5'- or 3'-termini of the molecules [21]. Importantly, multiple recent studies confirmed that isomiR expression profiles can deliver more insights as compared to canonical miRNAs profiles. Thus, in one recent report, the authors constructed a machine learning algorithm that allowed distinguishing thirty two different types of cancer based on the information about presence or absence of particular isomiRs but not canonical miRNAs [22]. It is also important to note that two 5'-isomiRs (i.e. isomiRs having different 5'-ends) of the same miRNA can have distinct set of target mRNAs due to the altered seed region [23, 24].

In this work we explored the landscape of ACE2 and TMPRSS2 regulation mediated by miRNAs and isomiRs in different human organs using bioinformatic analysis of publicly available paired miRNA/mRNA-sequencing datasets. We observed negative correlation of various miRNA and mRNA expressions across a number of samples, what together with the predicted miRNA-binding domains allowed identifying and quantifying putative miRNA-mediated targeting of certain mRNAs [25]. These results can facilitate developing novel therapeutic approaches for preventing coronavirus infections and contribute to the fundamental mechanisms of virus-host interactions.

## Results

### ACE2 and TMPRSS2 expression profiles across different organs

In order to examine expression profiles of ACE2 and TMPRSS2 in human tissues we used publicly available data from The Cancer Genome Atlas (TCGA; https://www.cancer.gov/tcga) which provides information on mRNA and miRNA expression in tumor as well as adjacent normal tissues for a variety of organs. Specifically, we collected data from 541 samples in total (11 different organs). We found that both enzymes were highly expressed in all analyzed tissues, see Fig 1 (numerical characteristics are listed in S1 Table). Thus, ACE2 showed the highest expression rate in the colon, kidney, stomach and liver, while in other organs its average expression level was 2-5 $\log_2$ units less and stayed at approximately the same level. The TMPRSS2 expression was higher as compared to ACE2 almost in all analyzed tissues (average $\log_2$(fold change) = 3.18) reaching its peak in the prostate.

### Landscape of interactions between isomiRs and ACE2 / TMPRSS2

To identify the putative interactions between isomiRs and ACE2 / TMPRSS2 genes in particular tissues we have applied the two-step procedure. Firstly, we used TargetScan software to generate a list of miRNAs that could potentially bind to the 3' UTR of target mRNAs. Next, we have performed a correlation analysis to identify highly expressed isomiRs having significant negative correlation with ACE2 / TMPRSS2 gene expression levels in each organ. As a result, we designated 10 and 23 isomiRs which could potentially target ACE2 and TMPRSS2,

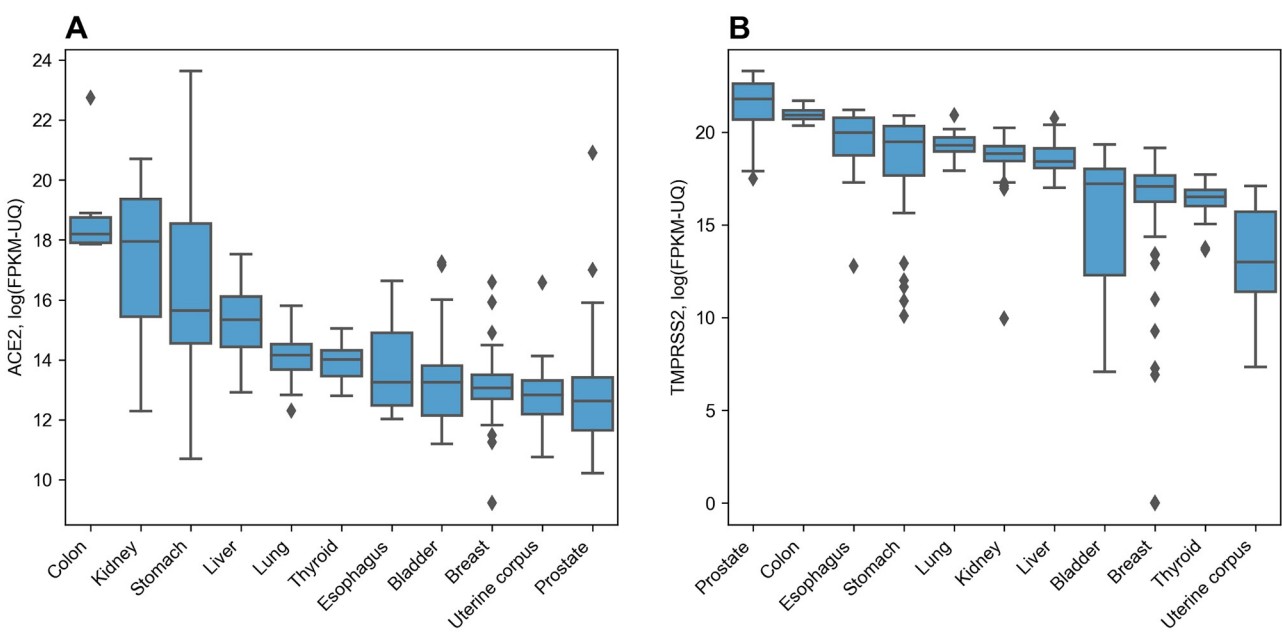

**Fig 1. Distribution of ACE2 and TMPRSS2 expression in human organs.** A: ACE2. B: TMPRSS2.

respectively. The highest number of ACE2 targeting isomiRs were found in kidney (6 isomiRs) and lung (3 isomiRs) tissues, while in the case of TMPRSS2 the highest number of regulating isomiRs were expressed in esophagus (11 isomiRs), stomach (8 isomiRs) and breast (6 isomiRs) cells (Fig 2, correlations and *p*-values are presented in S2 Table).

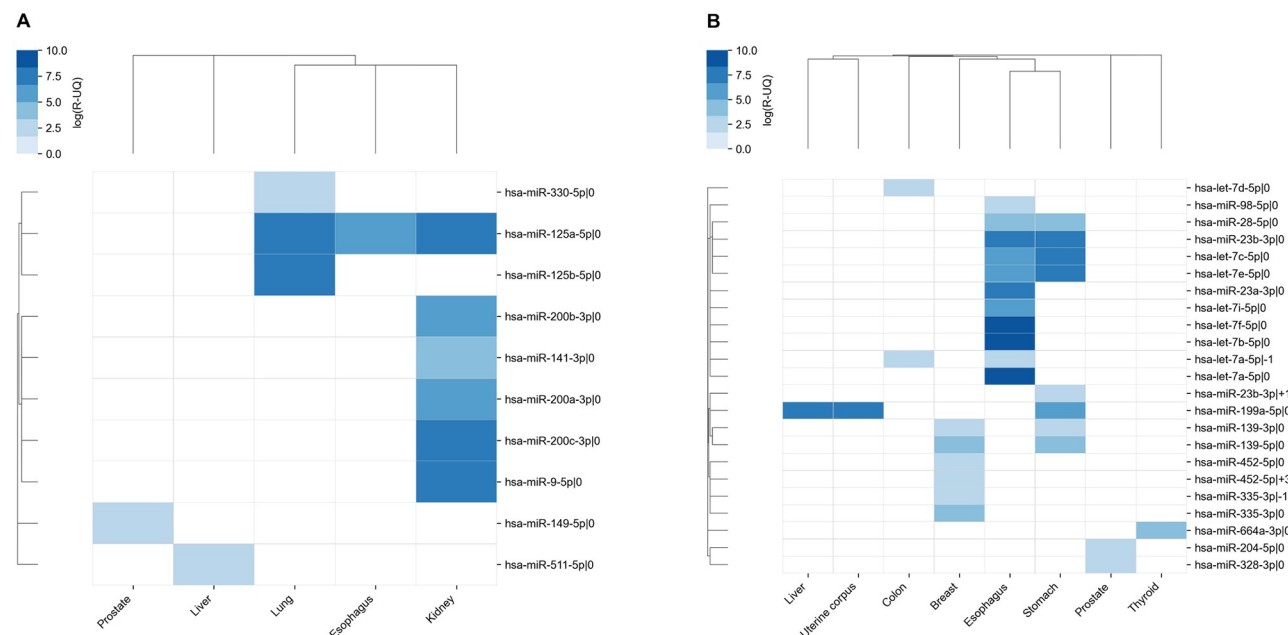

**Fig 2. IsomiRs regulating ACE2 and TMPRSS2.** A: ACE2. B: TMPRSS2. Color of each cell indicates the expression level of isomiR. A cell is empty (white) if isomiR is not targeting the corresponding gene.

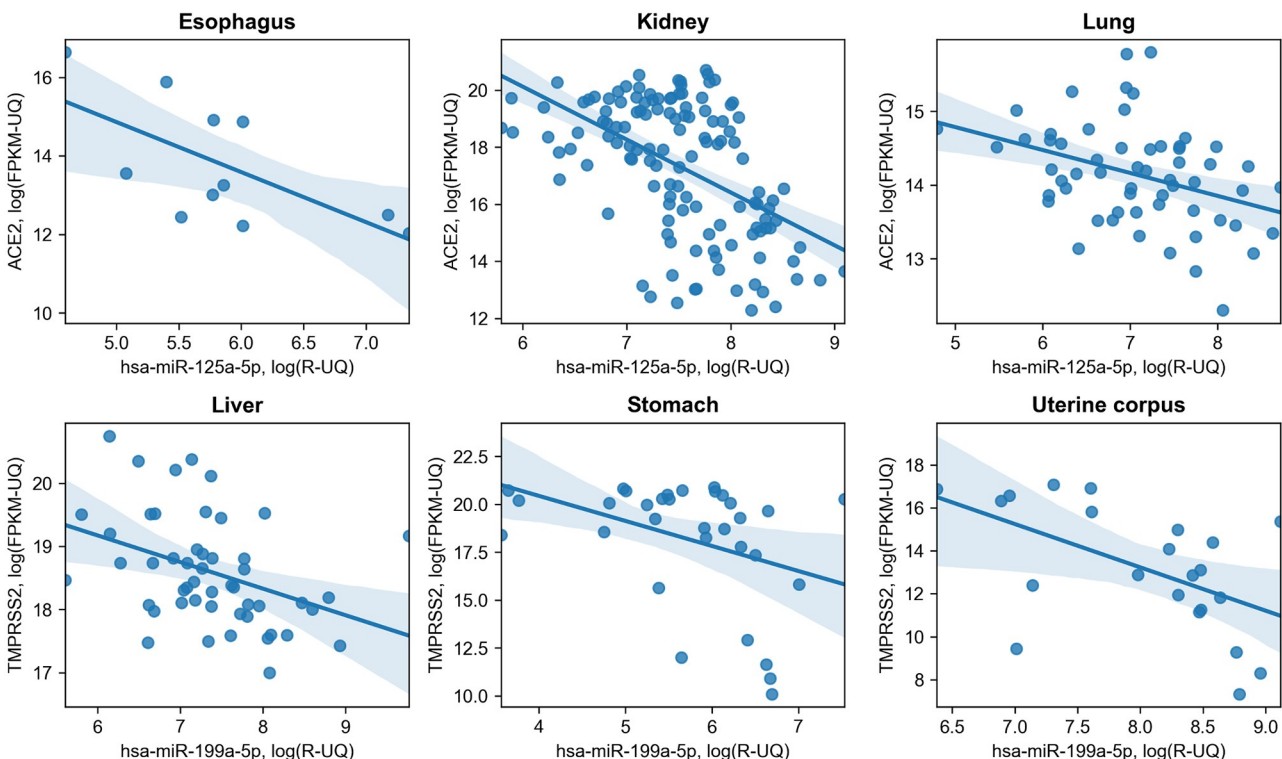

**Fig 3. Joint distribution of significant hsa-miR-125a-5p / ACE2 and hsa-miR-199-5p / TMPRSS2 interactions in different organs.** Light blue regions correspond to 95% confidence intervals.

The largest intersection corresponded to isomiRs regulating TMPRSS2 gene expression in esophagus and stomach (4 common isomiRs). Interestingly, some miRNAs (including hsa-miR-23b-3p, hsa-miR-335-3p and hsa-miR-452-5p) were expressed as both 5'-isomiRs (+1, -1 and +3 nucleotides respectively) and canonical forms. Furthermore, we identified two ACE2 / TMPRSS2 regulating miRNAs to be expressed in more than two tissues. The latter included hsa-miR-125a-5p (targeting ACE2 in esophagus, kidney and lung), as well as hsa-miR-199a-5p (regulating expression of TMPRSS2 in liver, stomach and uterine corpus, see Fig 3).

To further explore the interplay between different isomiRs and the corresponding mRNAs we have focused on genes hosting miRNAs in their introns. In total, we found 14 out of 33 isomiRs to be encoded within the introns in the sense orientation (see Table 1). Interestingly, hsa-miR-125a-5p (targeting ACE2 in lung, kidney and esophagus) and hsa-let-7e-5p (targeting TMPRSS2 in stomach and esophagus) were found to be encoded within the same intron of the SPACA6 gene.

## Lysine-specific demethylase 5B regulates ACE2 and TMPRSS2 through repression of let-7e / miR-125a and miR-141 / miR-200 activity

In order to determine the other players involved in the interplay between hsa-let-7e-5p / hsa-miR-125a-5p and ACE2 / TMPRSS2 genes we searched for proteins regulating the transcription of those miRNAs using literature-curated interaction databases. Thus, lysine-specific demethylase 5B (JARID1B, encoded by the KDM5B gene) had been previously shown to repress the transcription of hsa-let-7e / hsa-mir-125a miRNAs and miR-200 family (including miR-141, miR-200a, miR-200b, miR-200c and miR-429) by promoting H3K4me3 histone

**Table 1. Intronic sense pre-miRNAs and their host genes.**

| pre-miRNA | Host gene |
|---|---|
| hsa-mir-335 | MEST |
| hsa-mir-139 | PDE2A |
| hsa-let-7e, hsa-mir-125a | SPACA6 |
| hsa-mir-23b | AOPEP |
| hsa-let-7f-2, hsa-mir-98 | HUWE1 |
| hsa-mir-28 | LPP |
| hsa-miR-9-1 | C1orf61 |
| hsa-mir-149 | GPC1 |
| hsa-mir-204 | TRPM3 |
| hsa-miR-664a | RAB3GAP2 |

mark demethylation within the regulatory regions of miRNAs, thereby facilitating epigenetic repression of their transcription. The latter leads to the increase of ACE2 and TMPRSS2 genes expression since hsa-miR-125a-5p together with miRNAs from miR-200 family targets 3' UTR of ACE2 mRNA while hsa-let-7e-5p targets 3' UTR of the TMPRSS2. As a result, JARID1B epigenetic activity can indirectly regulate expression of ACE2 and TMPRSS2. The scheme of this interaction network is illustrated in Fig 4.

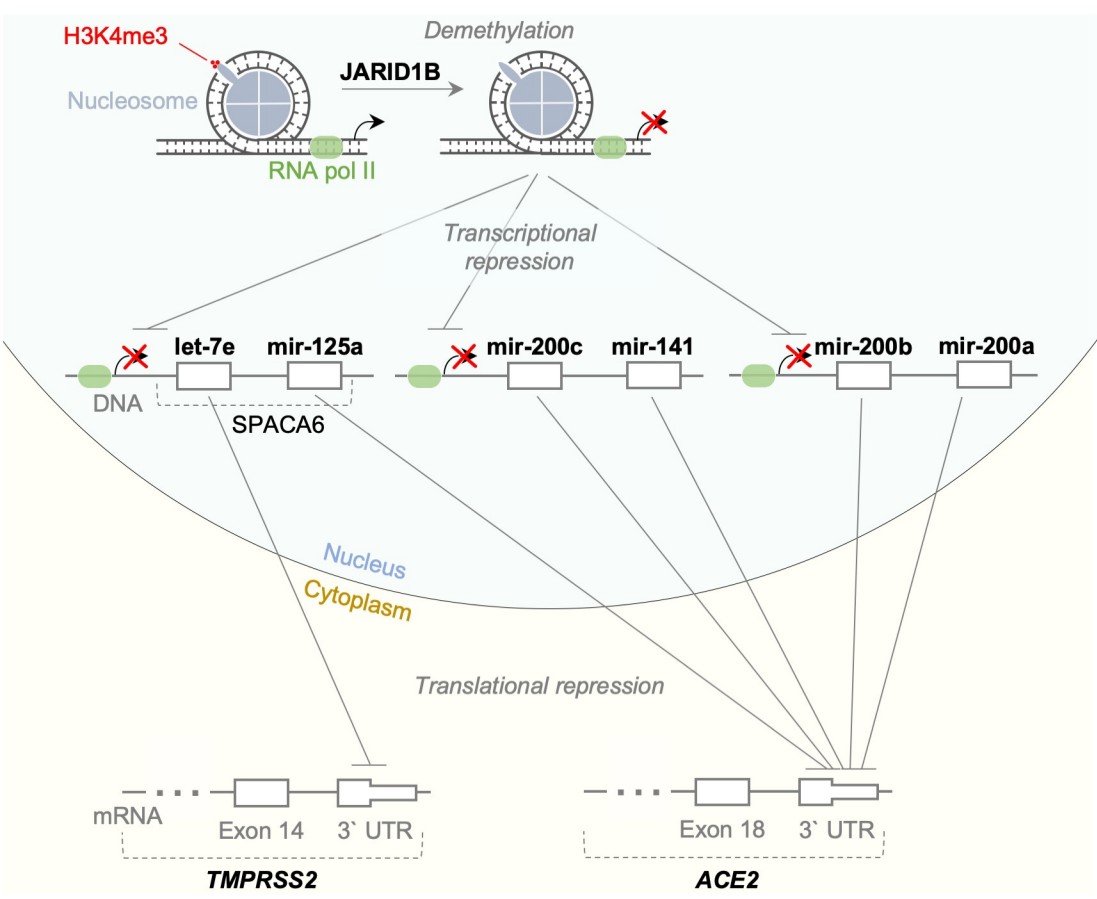

**Fig 4. Scheme of the interaction network consisting of JARID1B, let-7e / miR-125a and miR-141 / miR-200 miRNA families and ACE2 / TMPRSS2 enzymes.**

## Expression of JARID1B is neccesary for ACE2 and TMPRSS2 expression in the majority of human cells

To explore putative interactions between JARID1B and ACE2 / TMPRSS2 genes more deeply we analyzed publicly available single-cell RNA sequencing (scRNA-seq) data obtained from certain human organs (annotated and deposited by Sungnak et al. [26]). In general, the expression of all three genes was relatively low in most cell types which led to a high dropout rate. Specifically, the mean dropout rate (i.e. the ratio of the number of cells with no JARID1B, ACE2 or TMPRSS2 aligned reads to the total number of cells) across all cell types was 87.9% for KDM5B, 99.1% for ACE2 and 93.2% for TMPRSS2. Thus, a correlation between the expression of genes forming an arbitrary pair at a single-cell level is strongly biased due to a large fraction of cells with no reads.

To overcome the issue caused by high dropout rates we calculated the mean expression of each gene for all cell types from the explored datasets (see S3 Table). The latter was followed by a binarization of the obtained expression profiles. Specifically, a gene was considered "expressed" in a given cell type if its average expression in this cell type was greater than the first quartile of expressions across all cell types. As a result, the ACE2 gene was expressed in 114 out of 272 analyzed cells, while 100 cell types (87.7% of total) also expressed KDM5B. This observation indicates that the expression of KDM5B gene could be necessary for ACE2 expression (binomial test $p = 2.2 \times 10^{-7}$) in the majority of cells. A similar conclusion can be done for KDM5B and TMPRSS2 genes since 135 out of 159 cell types (84.9%) expressing TMPRSS2 also expressed KDM5B (binomial test $p = 1.61 \times 10^{-7}$).

Finally, we identified the cell types where expression levels of KDM5B, ACE2 and TMPRSS2 genes were higher than the corresponding upper quartiles (Table 2). Interestingly, the list of cell types having high expression levels of the aforementioned genes included: nasal cells (ciliated and secretory), bronchial cells (ciliated, secretory and basal) as well as ciliated and alveolar cells of lung parenchyma.

**Table 2. Cell types with high levels of KDM5B, ACE2 and TMPRSS2 expression.**

| Organ | Cell type | KDM5B | ACE2 | TMPRSS2 |
|---|---|---|---|---|
| Nasal | Ciliated | 0.85 (96.7%) | 0.12 (97.1%) | 0.74 (95.2%) |
| Nasal | Secretory | 0.92 (97.4%) | 0.13 (97.4%) | 0.52 (91.5%) |
| Lung parenchyma | Type 2 alveolar | 0.32 (87.5%) | 0.03 (90.1%) | 1.02 (97.8%) |
| Prostate | Club | 0.30 (86.0%) | 0.01 (79.0%) | 1.00 (97.4%) |
| Pancreas | Ductal | 0.52 (92.3%) | 0.01 (82.0%) | 0.65 (94.5%) |
| Bronchi | Secretory | 0.72 (94.5%) | 0.07 (95.2%) | 0.37 (88.2%) |
| Bronchi | Ciliated | 0.66 (93.8%) | 0.03 (90.4%) | 0.45 (90.1%) |
| Rectum | Progenitor | 0.26 (80.1%) | 0.02 (89.7%) | 0.69 (94.9%) |
| Testis | Spermatogonial stem cell | 0.81 (96.3%) | 0.04 (92.6%) | 0.08 (79.4%) |
| Lung parenchyma | Ciliated | 0.40 (89.3%) | 0.03 (91.2%) | 0.44 (89.7%) |
| Bronchi | Basal | 0.75 (95.2%) | 0.04 (92.3%) | 0.07 (78.3%) |
| Prostate | Hillock | 0.26 (80.5%) | 0.01 (77.2%) | 0.54 (92.3%) |
| Rectum | Enteriendocrine | 0.33 (87.9%) | 0.04 (93.8%) | 0.32 (86.8%) |
| Kidney | Epithelial progenitor cell | 0.24 (77.6%) | 0.02 (86.0%) | 0.25 (84.6%) |

Gene expression estimates are presented as $\log_2$-transformed counts per ten thousand mapped reads averaged across all cells of a given type. The values in brackets indicate the corresponding percentile of the expression distribution across all cell types. The rows of a table are sorted according to mean expression of KDM5B, ACE2 and TMPRSS2 genes.

## Discussion

In this work, we explored potential interactions between miRNAs and ACE2 / TMPRSS2 genes in various human organs. Along with the crucial role of both enzymes during SARS-CoV / SARS-CoV-2 entry into the cell, ACE2 and its targeting miRNAs have been also shown to affect the development of acute respiratory distress syndrome (ARDS). Thus, previous studies have indicated the critical role of ACE2 in acute lung injury, since ACE2 deficiency in the lungs enhanced the ARDS pathogenesis [27]. It was further shown that ACE2 gene becomes downregulated in the lungs during SARS-CoV infection [28], while another research team reported strong upregulation of hsa-miR-200c-3p / hsa-miR-141-3p cluster upon avian influenza virus H5N1 infection, presumably induced by viral proteins [29]. Interestingly, hsa-miR-200c-3p and hsa-miR-141-3p can directly target 3' UTR of ACE2 mRNA, since transfection of HEK293T cells with the corresponding miRNA mimics and inhibitors resulted in significant decrease or increase of ACE2 expression, respectively [29]. These results indicate the putative link between the viral infection and consecutive ARDS development caused by downregulation of ACE2 in the lung tissues. Furthermore, miRNA overexpression and knockdown experiments performed on HK-2 renal tubular epithelial cells indicated that hsa-miR-125b (which belongs to the same miRNA family as hsa-miR-125a) directly targets ACE2 mRNA [30].

Using the literature-curated interaction database we have found strong indications that the JARID1B gene can repress let-7e / miR-125a as well as miR-200 families presumably via epigenetic mechanisms. Thus, Mitra et al. [31] have previously provided experimental evidence that the JARID1B gene represses transcription of the let-7e and miR-125a by promoting H3K4me3 demethylation. Specifically, shRNA-mediated knockdown of JARID1B in MCF-7 and T47D cells led to a several-fold increase in expression of both miRNAs, while simultaneous ChIP analysis confirmed the H3K4 demethylation of the corresponding regulatory DNA sequences. In another report, Enkhbaatar et al. [32] showed that JARID1B represses transcription of miR-200 family by a similar mechanism. Specifically, the overexpression of JARID1B in lung cancer cell line A549 resulted in 3-fold decrease of miR-200a and miR-200c expression while JARID1B knockdown led to 1.5-fold increase of their steady-state levels. Subsequent ChIP assay showed putative changes in the H3K4me3 methylation of the corresponding DNA regions. Therefore, our results go in accordance with the experimental data showing the existence of regulatory network involving let-7e / miR-125a / miR-200 families, histone demethylase JARID1B as well as ACE2 / TMPRSS2, and further suggest a new way for regulating ACE2 expression. Furthermore, our single-cell RNA sequencing data analysis strongly supports the existence of such interactions in other cells, and indicates that in the majority of human cells ACE2 and TMPRSS2 are not expressed without JARID1B. In particular, high expression levels of JARID1B, ACE2 and TMPRSS2 in human respiratory epithelium cells intimate that further investigation of the identified regulatory network could expand our understanding of the viral infection pathogenesis.

The potential interactions between ACE2 and histone modifiers such as HAT1, HDAC2, and JARID1B in the lung has been previously shown by Pinto and co-authors [33]. Furthermore, Gordon D. et al. suggested the inhibition of histone deacetylase 2 (HDAC2) to be used as a promising strategy for COVID-19 therapy [34]. Specifically, the authors described high-confidence interaction between HDAC2 protein and the main viral protease Nsp5, supposing that Nsp5 may inhibit HDAC2 transport into the nucleus thereby altering its functional activity. In addition, they suggest testing certain HDAC2 repressors including Valproic acid and Apicidin since targeting HDAC2 can result in antiviral activity. Interestingly, Valproic acid was also shown to directly inhibit JARID1B in human embryonic kidney cells (HEK 293) [35] and H9 human embryonic stem cells (H9 hESC) [36]. Thus, repression of histone modifiers can adjust

the cell response to viral infection by engaging a complex regulatory mechanism involving ACE2 and TMPRSS2 genes. However, additional experiments are required to confirm the impact of the described network as well as drugs targeting the above mentioned enzymes.

Several miRNAs identified in this work have been previously reported to contribute to the viral infection process as well. Thus, nucleocapsid protein of the human coronavirus OC43 (HCoV-OC43) had been shown to bind to hsa-miR-9-5p, a prominent negative regulator of the transcription factor NF-$\kappa$B [37]. The latter results in NF-$\kappa$B activation and consequent alteration of the innate immune response. Finally, Mallick B. et al. [38] have shown that hsa-miR-98-5p targets 3' UTR of SARS-CoV Spike protein in bronchoalveolar stem cells. The authors further hypothesized that such interactions can be used by the virus to evade fast elimination by the immune system.

While several other miRNAs had been also reported to be aberrantly expressed during the coronavirus infection, they were not among the putative regulators of ACE2 or TMPRSS2 genes identified in this work. For instance, using deep sequencing of mouse lung miRNome upon SARS-CoV infection Peng and co-authors [39] showed that hsa-let-7f-5p, hsa-miR-139-3p, hsa-miR-139-5p were up-regulated directly after the infection in lung cells. A similar miRNA-seq experiment was performed on RNA isolated from Calu-3 human lung adenocarcinoma cells infected by MERS-CoV [40]. Four other miRNAs were aberrantly expressed, including down-regulated hsa-miR-98-5p as well as up-regulated hsa-miR-125a-5p, hsa-miR-125b-5p and hsa-miR-141-3p. Importantly, due to the structural similarity between MERS-CoV / SARS-CoV and SARS-CoV-2, we hypothesize that the described mRNA/miRNA interactions may contribute to maintenance of ACE2 / TMPRSS2 expression levels during coronavirus infection, thus regulating the infection process.

## Materials and methods

### Public RNA sequencing data

Paired RNA-seq / miRNA-seq data of the TCGA project was downloaded from GDC portal (https://portal.gdc.cancer.gov/) in *.FPKM-UQ.txt and *.mirbase21.isoforms.quantification.txt tables formats. The data included samples from several tissues: 19 for bladder (TCGA-BLCA), 104 for breast (TCGA-BRCA), 8 for colon (TCGA-COAD), 11 for esophagus (TCGA-ESCA), 127 for kidney (TCGA-KICH, TCGA-KIRC, TCGA-KIRP), 50 for liver (TCGA-LIHC), 58 for lung (TCGA-LUAD, TCGA-LUSC), 52 for prostate (TCGA-PRAD), 32 for stomach (TCGA-STAD), 58 for thyroid (TCGA-THCA) and 22 for uterine corpus (TCGA-UCEC). Fragments per kilobase of transcript per million mapped reads upper quartile (FPKM-UQ) scale [41] was used to quantify mRNA expression, while reads normalized by upper quartile (R-UQ) were used as an isomiR expression scale, all values were $\log_2$-transformed. The expression of a 5'-isomiRs was calculated as a combined (summed up) value of expression of all isomiRs having the same 5'-end genomic coordinate according to miRBase v21 [42]. The transcripts having zero coverage in more than half of the analyzed samples for each tissue type were discarded. IsomiR nomenclature from [43] was used: number after a vertical bar denotes offset from 5'-end in the direction from 5'- to 3'-end. For example, hsa-miR-21-5p|+1 denotes isomiR which differs from the canonical form by the absence of one nucleotide from the 5'-end of the molecule.

### IsomiR / miRNA analysis

TargetScan v7.2 software [44] was used to predict the isomiRs potentially targeting 3' UTR of ACE2 and TMPRSS2 genes. The entries having weighted context++ score below 0.8 quantile were removed from the downstream analysis. For each tissue type we calculated Spearman

correlation coefficient between the expression of ACE2 / TMPRSS2 genes and the isomiRs predicted by TargetScan which were highly expressed in considered samples (20% of isomiRs with the highest median expression). The -0.3 correlation coefficient threshold and 0.05 *p*-value threshold were applied on the obtained values. Hierarchical clustering of isomiRs across tissues was done using Jaccard metric.

The list of intragenic miRNAs and their host genes was constructed by the following procedure:

1. The list of miRNAs with their genomic coordinates (hsa.gff3 file from miRBase v21) was converted into BED format.

2. Human genome annotation was downloaded from Ensemble v97 (Homo_sapiens. GRCh38.97.chr.gff3 file for GRCh38.p12 version) [45]. This file was further split into gene and exon annotations. The resulting annotation files were converted into BED format.

3. The list of miRNAs was intersected with lists of genes and exons separately using bedtools intersect v2.26.0 [46]. Finally, exonic miRNAs were excluded from the intragenic miRNA list and the entries from different strands were filtered out from the resulting files.

TransmiR v2.0 [47] literature-curated database was used to extract information on protein-miRNA regulatory interactions.

## Single-cell RNA sequencing data analysis

Annotated scRNA-seq files were downloaded from https://www.covid19cellatlas.org/ in *. h5ad format, see S3 Table for the list of file names and putative organs. Raw count values were normalized by the sum of counts in each cell and scaled using $10^4$ factor. Obtained values were $\log_2$-transformed.

## Software utilized

All code was written in Python 3 programming language with extensive use of Pandas [48] and NumPy [49] modules. Statistical analysis was performed using the SciPy [50]. Single-cell RNA sequencing data was processed using the Scanpy [51]. Plots were constructed using the Matplotlib / Seaborn [52]. All used data and source codes have been made available on GitHub (https://github.com/s-a-nersisyan/miRNA_ACE2_TMPRSS2).

## Supporting information

**S1 Table. Statistical summary of ACE2 and TMPRSS2 distribution in different organs.** (XLSX)

**S2 Table. Correlation analysis of isomiR-ACE2/TMPRSS2 interactions.** (XLSX)

**S3 Table. Cell type level scRNA-seq expression data.** (XLSX)

## Author Contributions

**Conceptualization:** Stepan Nersisyan, Maxim Shkurnikov, Andrey Turchinovich, Evgeny Knyazev, Alexander Tonevitsky.

**Data curation:** Stepan Nersisyan, Maxim Shkurnikov.

**Formal analysis:** Stepan Nersisyan, Andrey Turchinovich.

**Investigation:** Stepan Nersisyan, Maxim Shkurnikov, Evgeny Knyazev, Alexander Tonevitsky.

**Methodology:** Stepan Nersisyan, Maxim Shkurnikov, Andrey Turchinovich.

**Software:** Stepan Nersisyan.

**Supervision:** Maxim Shkurnikov, Alexander Tonevitsky.

**Visualization:** Stepan Nersisyan.

**Writing – original draft:** Stepan Nersisyan, Andrey Turchinovich.

**Writing – review & editing:** Stepan Nersisyan, Maxim Shkurnikov, Andrey Turchinovich, Evgeny Knyazev, Alexander Tonevitsky.

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
