## [Decision Letter · Decision Letter 0]

27 May 2020

PONE-D-20-12168

Integrative analysis of miRNA and mRNA sequencing data reveals potential regulatory mechanisms of ACE2 and TMPRSS2

PLOS ONE

Dear Dr. Nersisyan,

Thank you for submitting your manuscript for review to PLoS ONE. Your manuscript has been reviewed by one expert and I have also personally evaluated your article. I agree with the comments raised by the reviewer and we feel that your study has merit, but is not suitable for publication as it currently stands. Therefore, my decision is "Major Revision”.

You must revise accordingly and explain your revisions in a covering letter if you wish for us to consider your paper further for publication. We invite you to submit a revised version of the manuscript that addresses the concerns raised by the reviewer. Please pay attention to all the reviewer suggestions and give them due consideration.

Specifically:

While the paper is well-written and the methodology used appears to be sound, the reviewer indicates that the work is mainly based on correlative analyses and would deserve at least one experimental validation among the different hypotheses. I agree with this criticism, in particular because the correlations found are at the tissue and not at the cell level. I thus feel that this experimental validation would strongly improve the manuscript. I can eventually propose an alternative to this main criticism if the authors are able to check at the single cell level at least some of the interesting correlations and hypotheses they have highlighted in their study. Since miRNA data sets are not available at the single cell level, the authors may use either expression of host gene in scRNA-seq database and/or use of miRNA FISH data when available.

We look forward to receiving your revised manuscript.

Kind regards,

Bernard Mari, Ph.D

Academic Editor

PLOS ONE

We note that one or more of the authors are employed by a commercial company: SciBerg e.Kfm

Reviewers' comments:

Reviewer's Responses to Questions

**Comments to the Author**

1. Is the manuscript technically sound, and do the data support the conclusions?

Reviewer #1: Yes

2. Has the statistical analysis been performed appropriately and rigorously? 

Reviewer #1: Yes

3. Have the authors made all data underlying the findings in their manuscript fully available?

Reviewer #1: Yes

4. Is the manuscript presented in an intelligible fashion and written in standard English?

Reviewer #1: Yes

5. Review Comments to the Author

Reviewer #1: The manuscript PONE-D-20-12168 entitled “Integrative analysis of miRNA and mRNA sequencing data reveals potential regulatory mechanisms of ACES2 and TMPRSS2” by Nersisyan et al. describes putative ACE2 and TMPRSS2 expression regulation networks mediated by various miRNA isoforms (isomiR). To do this, they used publicly available paired miRNA/mRNA-sequencing data from The Cancer Genome Atlas (TCGA) project across different human organs. They claim to have identified several miRNA families targeting ACE2 and TMPRSS2 genes in multiple tissues. Their salient discovery is that the lysine-specific demethylase 5B (JARID1B), encoded by the KDM5B gene, can indirectly affect ACE2 / TMPRSS2 expression by repressing transcription of hsa-let-7e / hsa-mir-125a and hsa-mir-141 /hsa-miR-200 miRNA families which are targeting these genes.

The paper is well-written. The methodology used appears to be sound and coherent. However, the paper is very limited in scope and only generates hypothesis that would require thorough experimental validations. This limitation is, in fact, acknowledged by the authors, as their title reads “potential regulatory mechanisms”.

Although some of the hypothesis are validated by some published work, the conclusions of the paper are in my view far too definitive considering the authors describes essentially correlative studies. They do not provide any validations. I feel that this experimental validation is an absolute requirement for publication as simple bibliographic argumentation can be largely biased.

6. PLOS authors have the option to publish the peer review history of their article (what does this mean?). If published, this will include your full peer review and any attached files.

Reviewer #1: No

---

## [Author Response · Author response to Decision Letter 0]

12 Jun 2020

Dear Dr. Mari,

Please find enclosed a revised version of our manuscript «Integrative analysis of miRNA and mRNA sequencing data reveals potential regulatory mechanisms of ACE2 and TMPRSS2». We would like to thank you and the reviewer for valuable comments and criticism. We agree with the reviewer that sequence-level target prediction followed by correlation analysis only generates a hypothesis on miRNA-gene interaction which should be further experimentally validated. However, our main finding, namely, indirect miRNA-induced interaction between JARID1B and ACE2 / TMPRSS2 is actually supported by several high-confidence experimental reports (non-correlative) highlighted in the existing literature. We have additionally  rewritten several paragraphs of the discussion accordingly to uncover these details.

Furthermore, we addressed the idea related to the single-cell level analysis raised by the Editor. Due to the absence of scRNA-seq miRNA data we analyzed co-expression of JARID1B, ACE2 and TMPRSS2 in multiple human organs using data deposited from https://www.covid19cellatlas.org/ . Our results suggest that ACE2 and TMPRSS2 cannot be well expressed without JARID1B in the majority of analyzed cells. Additionally, we found that three genes of our interest are highly expressed in human respiratory epithelium cells. We suppose that these findings improved our work and provided more evidence on the interactions discovered.

We hope that the revised version is now acceptable for publication in PLOS ONE.

Sincerely,

Stepan Nersisyan,

on behalf of all co-authors

---

## [Editor Report · Decision Letter 1]

18 Jun 2020

PONE-D-20-12168R1

Integrative analysis of miRNA and mRNA sequencing data reveals potential regulatory mechanisms of ACE2 and TMPRSS2

PLOS ONE

Dear Dr. Nersisyan,

Thank you for resubmitting your manuscript to PLOS ONE. While you have adequately addressed most of the queries in the review and that the revised manuscript is significantly improved from its original submission, some minor issues still needs to be addressed before full acceptance of the paper.

Specifically:

The authors now provide interesting data at the single cell regarding the cell types with high level of KDM5B, ACE2 and TMPRSS2 expression. They should specify in the new Table 2 the significance and the mode of calculation of these values. We also strongly encourage the authors to provide the code used in their IsomiR / miRNA analysis as a supplemental material.

We look forward to receiving your revised manuscript.

Kind regards,

Bernard Mari, Ph.D

Academic Editor

PLOS ONE

---

## [Author Response · Author response to Decision Letter 1]

23 Jun 2020

Dear Dr. Mari,

Thank you very much for valuable comments and suggestions, we have revised our manuscript accordingly. Specifically,

Point 1: "They should specify in the new Table 2 the significance and the mode of calculation of these values."

Response 1: we have added percentiles of the expression distribution in Table 2 to make the presented values more informative. Besides, we have rearranged the rows of the corresponding table according to their average expression. The detailed explanation was added under the table.

Point 2: "We also strongly encourage the authors to provide the code used in their IsomiR / miRNA analysis as a supplemental material."

Response 2: we have made all used source codes publicly available on GitHub according to your recommendation. The statement with the website link was added to the “Software utilized” section of the manuscript. 

Finally, we have corrected several stylistic / language issues in the manuscript text.

Thank you again for your time and consideration.

Sincerely,

Stepan Nersisyan,

on behalf of all co-authors

---

## [Editor Report · Decision Letter 2]

26 Jun 2020

Integrative analysis of miRNA and mRNA sequencing data reveals potential regulatory mechanisms of ACE2 and TMPRSS2

PONE-D-20-12168R2

Dear Dr. Nersisyan,

We’re pleased to inform you that your manuscript has been judged scientifically suitable for publication and will be formally accepted for publication once it meets all outstanding technical requirements.

Kind regards,

Bernard Mari, Ph.D

Academic Editor

PLOS ONE
---

## [Editor Report · Acceptance letter]

6 Jul 2020

PONE-D-20-12168R2 

Integrative analysis of miRNA and mRNA sequencing data reveals potential regulatory mechanisms of ACE2 and TMPRSS2 

Dear Dr. Nersisyan:

I'm pleased to inform you that your manuscript has been deemed suitable for publication in PLOS ONE. Congratulations! Your manuscript is now with our production department. 

Kind regards, 

on behalf of

Dr. Bernard Mari 

Academic Editor

PLOS ONE